# WildCAT3D: Appearance-Aware Multi-View Diffusion in the Wild

**Morris Alper**[1,2]* **David Novotny**[2] **Filippos Kokkinos**[2] **Hadar Averbuch-Elor**[3] **Tom Monnier**[2]

[1]Tel Aviv University          [2]Meta AI          [3]Cornell University

https://wildcat3d.github.io

**Training image collections** *in the wild*

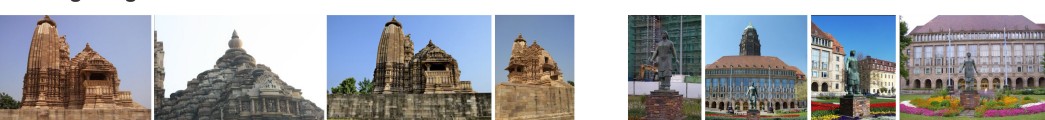

**Novel-view synthesis from a single image**

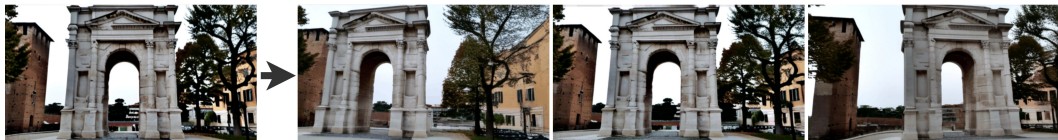

**Novel-view synthesis with appearance control**          "a spring day with a clear blue sky"

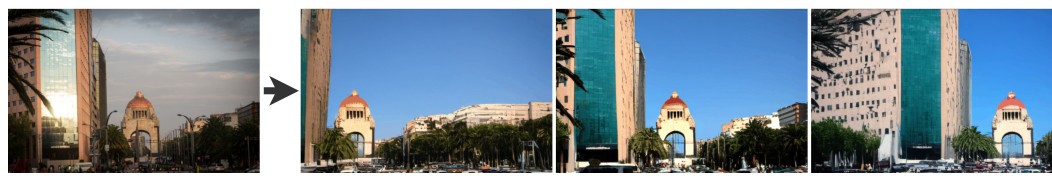

Figure 1: **WildCAT3D. (top)** We use large image collections captured *in the wild* to train our feed-forward novel-view synthesis model. **(middle)** At inference time, WildCAT3D can generate full scene-level novel views from a single image of a new (never-before encountered) scene. **(bottom)** It can also be used to control the appearance of the generated views, *e.g.* via a text prompt.

## Abstract

Despite recent advances in sparse novel view synthesis (NVS) applied to object-centric scenes, scene-level NVS remains a challenge. A central issue is the lack of available clean multi-view training data, beyond manually curated datasets with limited diversity, camera variation, or licensing issues. On the other hand, an abundance of diverse and permissively-licensed data exists in the wild, consisting of scenes with varying appearances (illuminations, transient occlusions, etc.) from sources such as tourist photos. To this end, we present WildCAT3D, a framework for generating novel views of scenes learned from diverse 2D scene image data captured in the wild. We unlock training on these data sources by explicitly modeling global appearance conditions in images, extending the state-of-the-art multi-view diffusion paradigm to learn from scene views of varying appearances. Our trained model generalizes to new scenes at inference time, enabling the generation of multiple consistent novel views. WildCAT3D provides state-of-the-art results on single-view NVS in object- and scene-level settings, while training on strictly fewer data sources than prior methods. Additionally, it enables novel applications by providing global appearance control during generation.

---

*Work done during Morris's internship at Meta.

39th Conference on Neural Information Processing Systems (NeurIPS 2025).

# 1 Introduction

Imagine observing the Golden Gate Bridge from afar at different viewpoints as the weather conditions change – in fog, bright sunlight, in rain and at night. From these observations, one can intuit the 3D structure of the Golden Gate Bridge, and can likely imagine how similar bridges might appear from various viewpoints. Our work applies a similar intuition to novel view synthesis (NVS) – the task of predicting a new 2D view of a scene which has been partially observed – to generate views of new scenes by learning from observations differing in global appearance. While recent progress in NVS has been achieved by leveraging powerful multi-view diffusion models, as popularized by CAT3D Gao et al. [2024], these models typically have poor applicability to full scenes due to scarce clean multi-view data. As such, these models typically build off pretrained image generation models which are fine-tuned on limited datasets of synthetic renderings or crowd-sourced videos capturing isolated objects, with poor applicability to full scenes like the MegaScenes collection [Tung et al., 2024]. On the other hand, such scenes are abundantly covered in permissively-licensed image collections captured *in the wild* from the Internet, in which scene views differ greatly in appearance (*e.g.*, aspect ratios, lighting, weather conditions or transient occlusions), making them incompatible with existing multi-view diffusion architectures. In this work, we unlock their ability to learn from this scalable source of readily-available, diverse, and permissively licensed scene data.

We propose WildCAT3D, a multi-view diffusion model *à la* CAT3D which can be learned from in-the-wild Internet data through appearance awareness. Our key insight is that inconsistent data can be leveraged during multi-view diffusion training to learn consistent generation, by specifically decoupling content and appearance when denoising novel views. More concretely, starting from the standard multi-view diffusion framework of CAT3D, we propose to explicitly integrate a feed-forward and generalizable appearance model whose goal is to capture the appearance properties of the input views. We do so by adding an appearance encoding branch to the model, designed to produce low-dimensional appearance embeddings that are used as an extra conditioning signal for the multi-view diffusion model. This branch is trained simultaneously with the diffusion model and a custom classifier-free guidance mechanism is applied to avoid oversaturation artifacts. During inference, the appearance embedding from the source view is injected to the target views, which allows us to preserve the appearance across the generated views. Intuitively, these design choices allow our model to "peek" at coarse appearance signals such as weather condition and aspect ratio, without leaking too much information about the target views to denoise.

In addition, in order to improve the viewpoint consistency, we augment our appearance-aware model by adapting a warp conditioning mechanism to the context of our multi-view diffusion framework. More specifically, for each target view to denoise, pixels from the source view are warped following the target viewpoint using a known depth map, and the resulting image is injected as an additional conditioning signal into the diffusion model. Intuitively, such a mechanism approximately indicates the correct placement of the scene, thus resolving the scale ambiguity that is inherent to the single-view NVS problem.

We exhaustively compare our method on standard NVS benchmarks and demonstrate superior performance, while training on fewer curated data sources than prior works. Importantly, this highlights the strength of our work in leveraging a larger set of samples from existing, permissively licensed imagery captured in-the-wild, rather than relying on heavily curated datasets. Our results also show strong performance on diverse scenes captured in tourist photos, including static video generation from single input frames and custom camera trajectories. In addition, our explicit modeling of appearance both allows us to learn from in-the-wild data to successfully generate consistent views of full scenes and also enables novel applications such as interpolation between views of differing appearances or NVS with appearance control via text, as showcased in Figure 1.

In summary, our key contributions are three-fold:

- A new appearance-aware multi-view diffusion model able to learn from in-the-wild images,
- Performance superior to state-of-the-art methods on single-view NVS benchmarks,
- Novel applications enabled by NVS with controlled appearances.

## 2 Related Work

**NVS with Diffusion Models.** Following recent progress in using diffusion models for generative modeling of image data, a line of works has successfully applied view-conditioned diffusion to NVS. Earlier works are limited to in-distribution object views with masked backgrounds and spherical camera poses [Watson et al., 2022, Zhou and Tulsiani, 2023]. The more recent Zero-1-to-3 [Liu et al., 2023a] and ZeroNVS [Sargent et al., 2023] train a diffusion model to generate a new view conditioned on an observed view and new camera pose, using curated multi-view image data as strong supervision to learn generalizable NVS. More recently, following works demonstrating diffusion models with a multi-view prior [Shi et al., 2023a, Li et al., 2023, Wang and Shi, 2023, Shi et al., 2023b, Yang et al., 2024a, Liu et al., 2023b], CAT3D [Gao et al., 2024] has shown SOTA NVS performance by leveraging this multi-view prior, allowing for multiple observed and/or output views to be processed in parallel. These approaches yield high-quality novel views, which may be used for tasks such as downstream 3D asset reconstruction. However, they are constrained by limited available training data, mostly covering single objects captured in crowd-sourced videos or synthetic renderings. Moreover, key sources of synthetic data may have contested licensing status. In contrast to these, our work enables training a multi-view diffusion model on a freely-licensed and abundant source of data in-the-wild, whose appearance variations are incompatible with these prior methods.

Another line of work leverages diffusion models with a warp-and-inpaint pipeline to enforce 3D consistency in unbounded scenes [Fridman et al., 2024, Yu et al., 2024a, Shriram et al., 2024, Chung et al., 2023, Yu et al., 2024b]. While this ensures 3D consistency, it often accumulates errors from depth estimation leading to inaccurate warps. By contrast, our method uses warps as a conditioning signal and not a strict constraint, allowing our model to correct such inaccuracies.

**NVS from in-the-Wild Image Data.** The abundance of Internet photo-tourism images has inspired research on extracting 3D structure from such data for tasks such as NVS. While such work predates modern neural methods [Snavely et al., 2006, Agarwal et al., 2011], recent works have used deep learning methods applied to large-scale photo collections which have been processed with SfM pipelines [Li and Snavely, 2018, Tung et al., 2024]. A number of works have concentrated on enhancing NVS and 3D reconstruction pipelines with explicit modeling of appearance variation between photos captured in-the-wild [Meshry et al., 2019, Li et al., 2020, Martin-Brualla et al., 2021, Chen et al., 2022, Kulhanek et al., 2024]. These methods perform test-time optimization on a single scene, with appearance representations extracted from pixel-level features or learned as directly optimized vectors. By contrast, our method trains a generalizable encoder which extracts appearance representations from image latents. Moreover, our framework uses these representations as conditioning signals for diffusion-based generation, which requires training- and inference-time adaptations to generate consistent, high-quality novel scene views. As such, our trained model is able to generalize to novel scenes without lengthy optimization and may directly use appearance information to condition high-quality 2D view generation, unlike existing works.

## 3 WildCAT3D

We proceed to define the WildCAT3D framework. We begin by providing background on the CAT3D [Gao et al., 2024] framework (Section 3.1) which our method extends. We then describe WildCAT3D's explicit modeling of appearance conditions (Section 3.2) and its scene scale disambiguation via warp conditioning (Section 3.3). Our full pipeline is illustrated in Figure 2.

### 3.1 Background: CAT3D

CAT3D is a multi-view diffusion model, adapted and fine-tuned from a text-to-image Latent Diffusion Model (LDM) [Rombach et al., 2022] in order to generate multiple views of a scene conditioned on source view(s) and source and target camera poses. Denoting the latent dimension as $k$ and the spatial resolution of latents as $n \times n$, the input noise (originally $k \times n \times n$) is first expanded to accept $v = 8$ slots corresponding to observed or unobserved views. Then, the input $I$ of shape $v \times (k + 7) \times n \times n$ consists of ground-truth latent for observed views and noise for unobserved views, concatenated channel-wise with 7 additional channels including a binary mask for observed and unobserved views (copied over all $n \times n$ spatial locations) and a $6 \times n \times n$-dimensional Plücker raymap [Sajjadi et al., 2022, Zhang et al., 2024] parametrizing the cameras for each view. To process these inputs, the

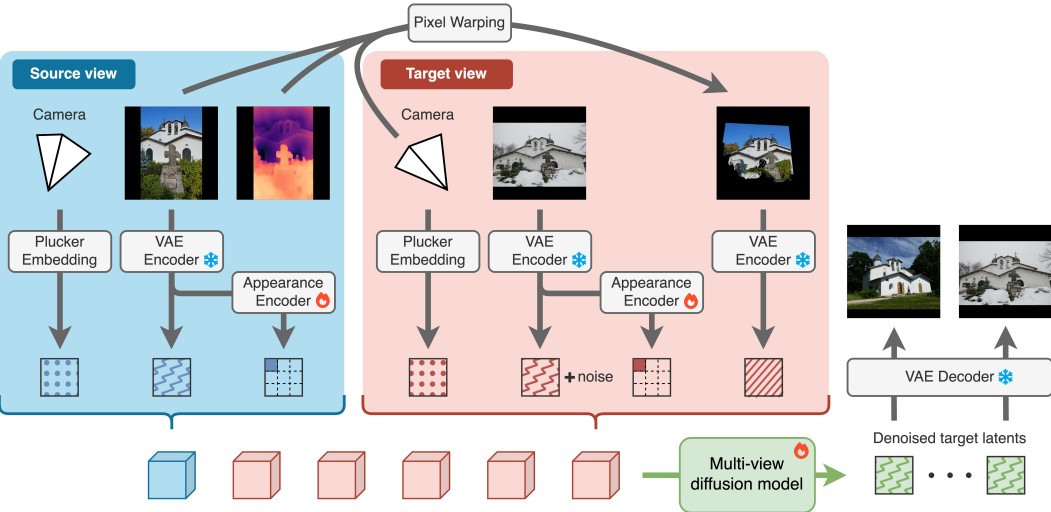

Figure 2: **Overview**. WildCAT3D learns to synthesize novel views by denoising target views of inconsistent appearances from a source view. Given a batch of source (blue) and target (red) views, we first compute camera embeddings and VAE latents. The latter are then fed to an encoder computing a small appearance vector copied across spatial locations, allowing the model to "peek" at appearance conditions. Finally, for target views, we compute additional warping embeddings using the VAE applied to warped source images calculated from an estimated depth map. These signals are channel-wise concatenated and fed to the diffusion model. During training (depicted above), noise is added to target view latents. During inference, target view latents are replaced by random noise while their appearance channels are copied from the source view branch.

LDM's self-attention layers are expanded to 3D attention, by selecting queries, keys, and values from all $v$ views. The original text-based cross-attention layers are removed (discarding the text encoder). At each denoising pass, ground-truth clean latents are passed in observed positions of the input, and the denoising loss in training is applied to slots for unobserved views only. CAT3D may be trained with any number ($\leq v$) of observed and unobserved views, integrating single- and multiple-view NVS in a single model. During inference, CAT3D generates multiple novel scene views in parallel.

This may be formulated mathematically as follows. A generative model for single images estimates the probability distribution $p(\mathbf{I}|t)$ of images $\mathbf{I}$ conditioned on some conditioning signal $t$. CAT3D models the distribution $p(\mathbf{I}^u|\mathbf{I}^o, \mathbf{c}^a)$ where $\cdot^{o,u,a}$ indicate observed/unobserved/all views, and $\mathbf{c}$ the camera information for a given view. This is parametrized by model weights $\theta$ and Gaussian noise $\mathbf{z}^u \sim (N(0,1))^u$ (i.i.d. noise for each unobserved view), and images are parametrized by the VAE.

### 3.2 Appearance-Aware Multi-View Diffusion

In-the-wild photo collections show appearance variations such as differing aspect ratios, lighting conditions, seasonal weather, transient occlusions, etc., and naively training CAT3D on such data results in similar inconsistencies at inference time (shown in our ablations). We thus propose to explicitly model appearance variations while encouraging consistency at inference time as follows.

**Generalizable Appearance Encoder.** We augment CAT3D with a feed-forward appearance encoder module, implemented as a shallow convolutional network applied to image latents. By compressing each view into a single, low-dimensional vector, this serves as an information bottleneck, encoding coarse global appearance variations without the capacity to leak fine-grained details in images. During training, the $d$-dimensional appearance embedding of each view is copied across each $n \times n$ spatial position. The resulting $k \times d \times n \times n$ tensor is concatenated to the input channels of CAT3D, allowing the model to "peek" at global appearances of both observed and unobserved views during training (despite unobserved views' latents themselves being noised). The appearance encoder is jointly trained with the model's denoising objective at train time; it is *generalizable* as it can be applied to new scenes at inference (unlike test-time methods such as Chen et al. [2022]).

Using the notation from Section 3.1, we formulate this approach as follows. We assume that each image $\mathbf{I}$ possesses a latent appearance variable $\mathbf{a}$, representing the conceptual space of appearance variations for the same underlying scene. WildCAT3D models the conditional distribution $p(\mathbf{I}^u|\mathbf{I}^o, \mathbf{c}^a, \mathbf{a}^a, \mathbf{w}^o)$, parametrized by model weights $\theta$, Gaussian noise $\mathbf{z}^u \sim (N(0,1))^u$, appearance variables $\mathbf{a}^a$, and warp information $\mathbf{w}^o$ (see Section 3.3).

We also assume that appearance can be derived given an image; as such, we learn encoder $A_\phi$ with weights $\phi$ that parametrizes $\mathbf{a} \approx A_\phi(\mathbf{I})$. Thus we model $p(\mathbf{I}^u|\mathbf{I}^o, \mathbf{c}^a, A_\phi(\mathbf{I}^a), \mathbf{w}^o)$, with the predicted distribution determined by weights $\theta, \phi$. We implement $A$ with a light-weight convolutional network to introduce an inductive bias towards global appearance. Note that $A_\phi$ is a lossy (bottleneck) function, compressing an image into a single, low-dimensional vector. Therefore, $I^u$ cannot be directly reconstructed from $A_\phi(\mathbf{I}^a)$ and thus modeling this distribution is non-trivial.

We note that this derives appearance representations directly from images, rather than using accompanying textual metadata. This has several advantages: Images may not be accompanied by descriptive text, conditioning on text would require additional novel components to bridge text and visual appearance spaces, and important appearance details may be poorly reflected in text, such as precise image aspect ratios and lighting conditions.

Our encoder design follows standard convolutional principles with gradually decreasing spatial dimensions to create a low-dimensional appearance bottleneck. The architecture requires low-dimensional output to serve as an effective information bottleneck. Overall, this uses a minimal network avoiding additional design choices or parameters that would require computationally intensive testing. Further details are provided in the appendix.

**Appearance-Aware Conditioning.** Following CAT3D, we apply classifier-free guidance (CFG) [Ho and Salimans, 2022] to achieve high visual quality; this drops conditioning signals (latents and camera rays for observed views) for unconditional training, and extrapolating between conditional and unconditional predictions in inference. However, further naively applying CFG to the appearance conditions (i.e. masking $\mathbf{a}^a$ in unconditional training ) results in oversaturation artifacts. We hypothesize that this is because appearance embeddings are tied to image lighting and color balance, and CFG is known to be prone to oversaturation when applied to components controlling "gain" of intensity values in images [Sadat et al., 2024]. Therefore we propose an appearance-aware conditioning method to achieve the benefits of CFG without such artifacts. We adapt the "unconditional" setting of standard CFG by keeping appearance conditions while dropping other observed view conditions: $p^{(\text{uncond})}(\mathbf{I}^u|\mathbf{c}^u, A_\phi(\mathbf{I}^a))$, i.e. we still condition on all appearance embeddings $A_\phi(\mathbf{I}^a) = (\mathbf{a}^o, \mathbf{a}^u)$ which includes those of observed views. In other words, WildCAT3D "peeks" at the global appearances of all views in both conditional and unconditional training settings of CFG.

**Appearance-Conditioned Inference** There is an inherent gap between the training objective of WildCAT3D (denoising views of varying appearances) and its use during inference (generating views fully consistent with an input view). To produce outputs that are consistent despite training on inconsistent data, we select the first observed view $\mathbf{I}_0$, calculate its appearance embedding $\mathbf{a}_0 = A_\phi(\mathbf{I}_0)$, and copy it into the appearance embedding channel of each unobserved view, then generating using appearance-aware CFG as described above. Intuitively, this conditions the generated views on the same overall appearance as the first observed view, and our results show that this indeed succeeds in matching its appearance characteristics (lighting, style, etc.). Interestingly, this solution also enables the preservation of the aspect ratio, resulting in generated views that are consistent enough to generate smooth and appealing videos given a single image.

Since the appearance embedding used at inference is arbitrary, we can actually use a completely external image to compute the desired appearance embedding and thus perform appearance transfer or appearance-controlled generation. In order to support appearance control through text, we propose to concatenate our model with a text-to-image retrieval model. More specifically, given a text prompt, we use CLIP Radford et al. [2021] to compute its similarity with pre-computed image embeddings from a given database. In practice, we perform this retrieval over approximately 10K images from MegaScenes, covering varying weather and lighting conditions similar to the full dataset.

## 3.3 Warp Conditioning

A central challenge for single-view NVS systems is the inherent scale ambiguity of single image input: given an observed view and the relative camera pose of a desired unobserved view, the scale

of the translation vector between the cameras' extrinsic poses is unknown [Ranftl et al., 2020, Yin et al., 2022]. We resolve this via the observation from [Tung et al., 2024] that scene scale can be injected by an image warped according to its depth aligned to the extrinsic camera scale. To adapt this to our multi-view diffusion setting, we warp the first observed view to the camera of each of the $v$ slots of CAT3D using an estimated depth map aligned to the scene's SfM pointcloud, concatenating the VAE latents of each warp as additional conditioning channels. Note that this is fully compatible with in-the-wild data since warps encode the correct pose even when differing in global appearance from target views. We show in our ablations that this warp conditioning is necessary for accurate viewpoint consistency.

To summarize, in total WildCAT3D has input of shape $v \times (2k + d + 7) \times n \times n$, where the $k + 7$ input channels of CAT3D ($k$ latents, 6 camera embeddings, 1 binary mask) are extended with $d$ channels for appearance embeddings and $k$ channels for warp latents.

### 3.4 Implementation Details

For WildCAT3D's pretrained image generative backbone, we use an open-source LDM similar to [Rombach et al., 2022]. We first expand it to a CAT3D model and train on the standard curated CO3D [Reizenstein et al., 2021] and Re10K [Zhou et al., 2018] multi-view datasets; we then expand to a full WildCAT3D model and fine-tune on MegaScenes [Tung et al., 2024] and CO3D. Our appearance module is a fully-convolutional network applied to image latents, outputting an embedding of dimension $d = 8$ for each image. This is copied to every spatial location and concatenated as $a$ additional channels to the denoiser network's input at each denoising step. By default we use $v = 8$ (slots for views), training with one observed and seven unobserved randomly-selected scene views. For video results (supp. mat.), we increase this to $v = 16$ slots at inference.

The additional input channels are handled through a lightweight 1×1 convolution projection layer, making them compatible with the lower input dimension expected by the LDM's denoising network while adding negligible extra parameters. See the appendix for further details.

To calculate warp images for warp conditioning, we unproject the first observed view to a 3D point cloud, and then render it from each camera pose. Following Tung et al. [2024], we unproject using depth calculated with DepthAnything [Yang et al., 2024b] and aligned with RANSAC to the COLMAP point cloud (corresponding to the cameras' extrinsic scale). We render warps by rendering this point cloud from each view, with points' RGB values derived from the first view.

## 4 Experiments

### 4.1 Novel View Synthesis Results

NVS metrics are provided in Table 1, comparing to the recent SOTA MegaScenes NVS model (MS NVS) along with prior models reproduced from [Tung et al., 2024]. Our method trained on MegaScenes directly (unlike the aggressive filtering used to train MS NVS) mostly achieves superior performance across the board, on both reference-based and generative metrics, and on out-of-distribution datasets (object-centric DTU and scene-centric Mip-NeRF 360). This is despite MS NVS and other prior models being trained on additional sources of multi-view data (see supp.). Visual comparisons are provided in Figures 3–6a and in our supmat, showing that WildCAT3D generally maintains consistency with observed views while hallucinating plausible content for unseen regions.

On the MegaScenes benchmark itself, our quantitative results are comparable to the MS NVS baseline. However, we observe a significant qualitative improvement when applied to views with novel trajectories, as seen in Figure 3. We suspect this reflects the MegaScenes baseline model being trained on image pairs selected using the same filtering method used to construct the MegaScenes test set, which consists of (ground truth, target) image pairs, while our method is not exposed to such filtering at train time. Hence, the baseline model may be partially overfit to the data format of this benchmark. Consistent with this, our model shows significantly better metrics on out-of-distribution NVS benchmarks including the challenging scene-level Mip-NeRF 360 benchmark (Table 1).

| Method | DTU [Jensen et al., 2014] | | | | | Mip-NeRF 360 [Barron et al., 2022] | | | | |
|---|---|---|---|---|---|---|---|---|---|---|
| | PSNR ↑ | SSIM ↑ | LPIPS ↓ | FID ↓ | KID* ↓ | PSNR ↑ | SSIM ↑ | LPIPS ↓ | FID ↓ | KID* ↓ |
| Zero-1-to-3 (released) | 6.872 | 0.210 | 0.565 | 128.9 | 0.297 | 10.72 | 0.287 | 0.526 | 171.2 | 1.126 |
| ZeroNVS (released) | 5.799 | 0.111 | 0.648 | 160.0 | 0.352 | 6.999 | 0.124 | 0.669 | 137.0 | 0.537 |
| Zero-1-to-3 (MS) | 7.637 | 0.276 | 0.516 | 101.9 | 0.223 | 12.92 | 0.383 | 0.443 | 67.65 | 0.163 |
| ZeroNVS (MS) | 8.019 | 0.307 | 0.483 | 87.41 | 0.158 | 13.78 | 0.412 | 0.406 | 60.68 | 0.139 |
| SD-Inpaint | 9.946 | 0.369 | 0.495 | 214.4 | 1.067 | 12.92 | 0.400 | 0.456 | 150.1 | 0.792 |
| MS NVS | 8.795 | 0.393 | 0.400 | 85.96 | 0.163 | 14.06 | 0.441 | 0.381 | 64.41 | 0.142 |
| **WildCAT3D (Ours)** | **10.77** | **0.426** | **0.388** | **57.32** | **0.039** | **14.77** | **0.445** | **0.352** | **42.17** | **0.050** |

| Method | Re10K [Zhou et al., 2018] | | | | | MegaScenes [Tung et al., 2024] | | | | |
|---|---|---|---|---|---|---|---|---|---|---|
| | PSNR ↑ | SSIM ↑ | LPIPS ↓ | FID ↓ | KID* ↓ | PSNR ↑ | SSIM ↑ | LPIPS ↓ | FID ↓ | KID* ↓ |
| Zero-1-to-3 (released) | 11.63 | 0.438 | 0.405 | 160.2 | 0.725 | 9.090 | 0.241 | 0.548 | 86.89 | 0.634 |
| ZeroNVS (released) | 9.487 | 0.353 | 0.456 | 123.0 | 0.352 | 7.471 | 0.151 | 0.616 | 69.10 | 0.487 |
| Zero-1-to-3 (MS) | 14.64 | 0.570 | 0.272 | 68.91 | 0.024 | 12.16 | 0.367 | 0.429 | **9.784** | 0.023 |
| ZeroNVS (MS) | 16.02 | 0.630 | 0.205 | 61.12 | 0.024 | 12.90 | 0.401 | 0.386 | 9.838 | 0.024 |
| SD-Inpaint | 15.54 | 0.643 | 0.269 | 118.9 | 0.396 | 12.36 | 0.392 | 0.425 | 38.48 | 0.242 |
| MS NVS | 17.22 | 0.666 | 0.177 | 60.01 | 0.023 | 13.40 | **0.445** | **0.344** | 11.58 | 0.040 |
| **WildCAT3D (Ours)** | **21.58** | **0.758** | **0.131** | **24.70** | **-0.001** | **13.92** | 0.439 | 0.355 | 9.871 | **0.015** |

Table 1: **Novel-view synthesis benchmarks.** We evaluate the single-view setup and compare our results to prior works as reported by Tung et al. [2024]: Zero-1-to-3 [Liu et al., 2023a], ZeroNVS Sargent et al. [2023], a naive inpainting method dubbed SD-Inpaint, and MegaScenes NVS model Tung et al. [2024]). Our method achieves superior performance compared to baseline methods while using strictly fewer data sources and not requiring aggressive data filtering. KID* indicates KID values multiplied by ten for readability. Best results are in **bold**.

## 4.2 Additional Applications

In Figures 4–5, we illustrate additional applications of WildCAT3D. By injecting the appearance embedding of an external image during inference, we may generate novel views of a scene while editing its appearance (e.g. generating a nighttime scene observed from a daytime photo). By concatenating this with CLIP text-to-image retrieval [Radford et al., 2021], we may use text to control the edit (e.g. "sunset"); as seen in the figure, text-based retrieval may effectively find images capturing the desired overall appearance for subsequent injection. Finally, by fine-tuning WildCAT3D with two observed input views, we can interpolate between views of a scene with differing appearances to produce a static video with a consistent appearance starting and ending at the two respective poses. At inference time, this uses camera poses along an interpolated trajectory between the two input cameras, along with the appearance of either the start or the end pose injected into each slot.

## 4.3 Analysis of Appearance Embeddings

To further interpret our results, we analyze the appearance embeddings produced by WildCAT3D's trained appearance encoder module. We cluster embeddings of approximately 20K MegaScenes (val and test) images with K-Means ($k = 100$). Figure 7 illustrates such clusters projected into two dimensions via PCA, and random exemplars from different clusters. Clusters contain images with similar aspect ratios, lighting conditions, and other global appearance factors (e.g. indoor vs. blue sky vs. nighttime), providing interpretability to the appearance encoder's functionality.

## 4.4 Ablations

We ablate key elements of our framework in Table 2 and Figure 6b. Removing warp conditioning leads to spatial misalignments due to scene scale ambiguity. Further removing appearance modeling, *i.e.* fine-tuning CAT3D directly on in-the-wild data, leads to appearance inconsistencies. By contrast, our full model improves on all benchmarks except Re10K from training on in-the-wild data over the base CAT3D, due to our careful modeling of appearance and scene scale. Note that Re10K has very limited diversity and camera movement, and is in-distribution for the base CAT3D; we thus interpret the stronger performances of the latter on Re10K as an overfitting effect.

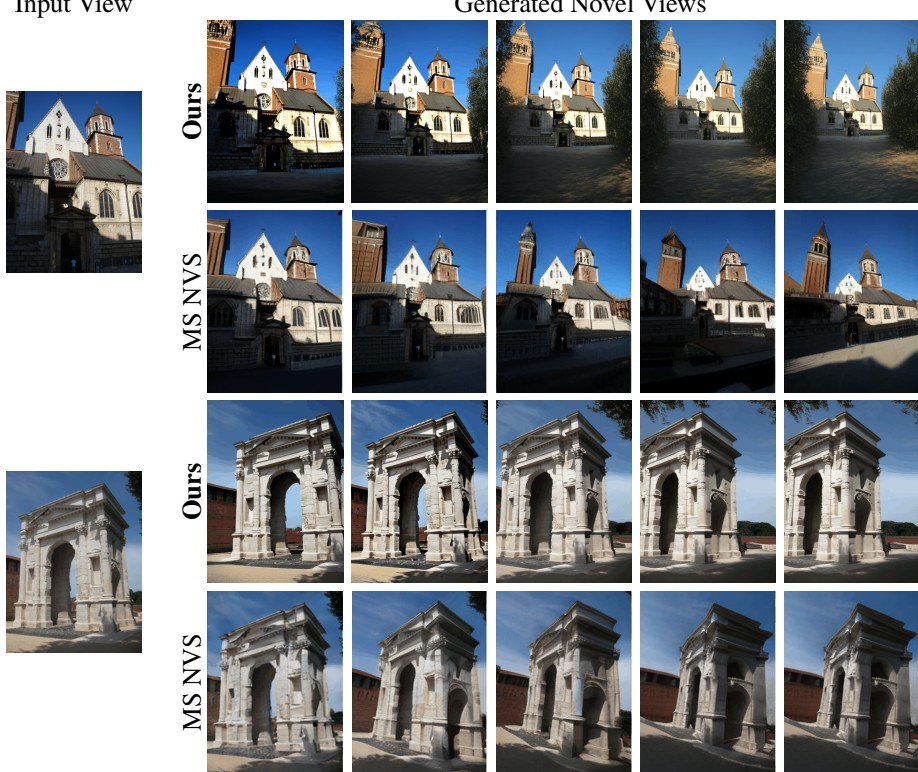

Figure 3: **Qualitative comparison on MegaScenes with novel trajectories.** Using single images as input (left), we show results for WildCAT3D and MegaScenes NVS model (MS NVS) on scenes unseen during training, conditioned on a continuous camera trajectory. We see that our model significantly outperforms prior SOTA at generating consistent and high-quality sequences from single views. We encourage the reader to check our video results in our supmat to further assess the quality gap.

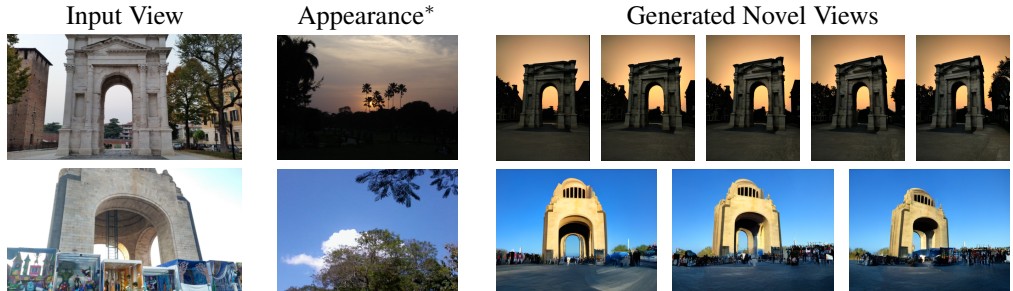

Figure 4: **Application: appearance-controlled generation.** Starting from a source view (left) and an additional image with a specific appearance (middle), our model is able to synthesize novel views that are not only consistent with the source view content and the desired viewpoints, but also consistent with the appearance style of the additional image (right). We perform text-guidance by concatenating our model with a text-to-image retrieval model. *Retrieved with text prompts ''`sunset`'' and ''`a spring day with a clear blue sky`'' respectively.

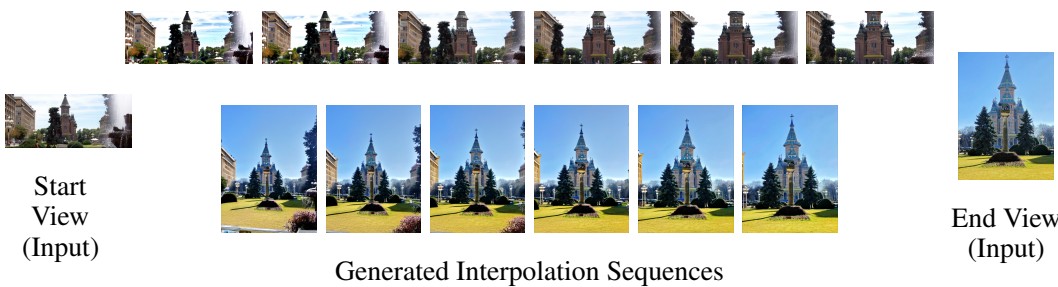

Start View (Input)

Generated Interpolation Sequences

End View (Input)

Figure 5: **Application: in-the-wild interpolation.** When fine-tuned with two observed views, WildCAT3D can interpolate between scene views with differing appearances. Injecting the appearance embedding of either the start or the end pose yields generated views with consistent appearances.

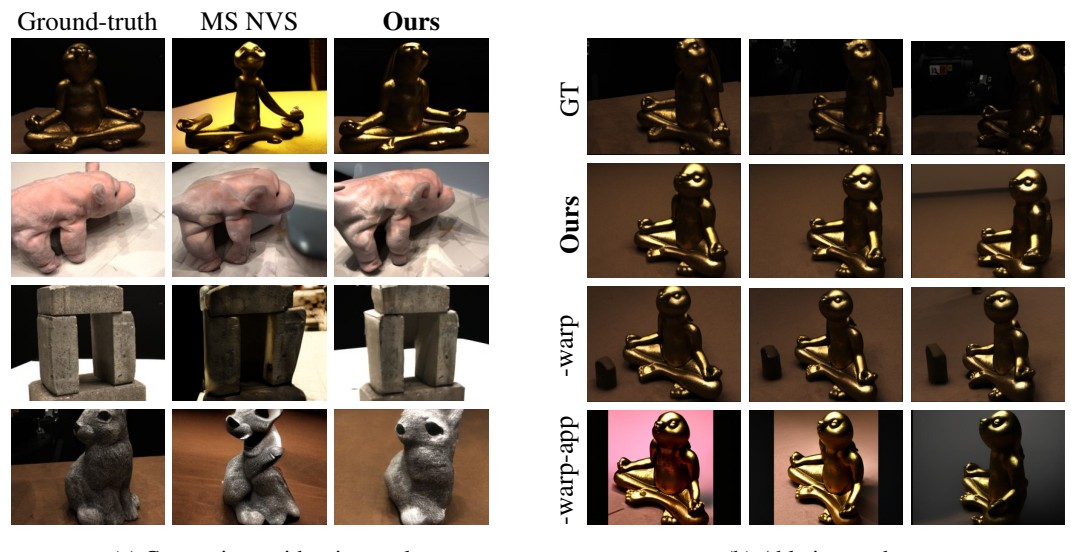

(a) Comparison with prior work

(b) Ablation study

Figure 6: **Qualitative NVS results.** **(a)** We compare predicted views to the ground-truth for MegaScenes NVS model from Tung et al. [2024] (MS NVS) and ours. Our model shows greater consistency with target poses as well as better visual quality, consistent with our quantitative results. **(b)** Removing warp conditioning (-warp) results in misalignment relative to ground-truth camera poses. Training CAT3D directly on in-the-wild data (*i.e.* without warps and appearance embeddings, -warp-app) yields inconsistent output images.

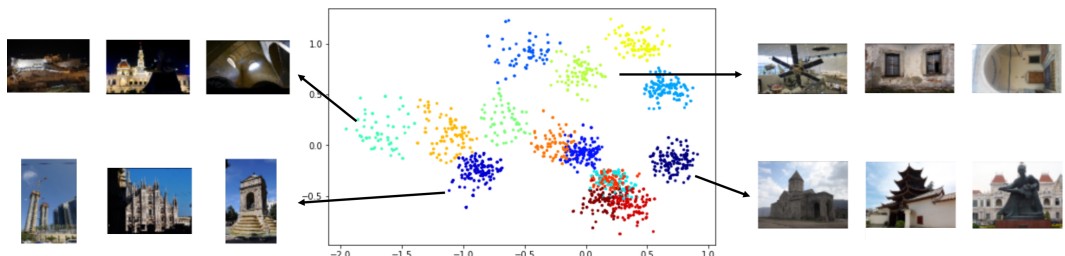

Figure 7: **Appearance embedding analysis.** A subset of K-Means clusters are visualized with a 2D PCA and random cluster images. They tend to show similarities in appearance and aspect ratio.

| Method | DTU [Jensen et al., 2014] | | | | | Mip-NeRF 360 [Barron et al., 2022] | | | | |
| | PSNR ↑ | SSIM ↑ | LPIPS ↓ | FID ↓ | KID[*] ↓ | PSNR ↑ | SSIM ↑ | LPIPS ↓ | FID ↓ | KID[*] ↓ |
|---|---|---|---|---|---|---|---|---|---|---|
| **WildCAT3D (Ours)** | **10.77** | **0.426** | **0.388** | 57.32 | **0.039** | **14.77** | **0.445** | **0.352** | **42.17** | **0.050** |
| -warp | 9.795 | 0.374 | 0.439 | 54.89 | 0.042 | 13.98 | 0.404 | 0.395 | 44.82 | 0.056 |
| -warp-app | 8.699 | 0.281 | 0.491 | 68.80 | 0.069 | 13.90 | 0.390 | 0.417 | 46.80 | 0.064 |
| Base CAT3D | 10.25 | 0.399 | 0.424 | **53.96** | 0.042 | 13.92 | 0.399 | 0.410 | 57.96 | 0.205 |

| Method | Re10K [Zhou et al., 2018] | | | | | MegaScenes [Tung et al., 2024] | | | | |
| | PSNR ↑ | SSIM ↑ | LPIPS ↓ | FID ↓ | KID[*] ↓ | PSNR ↑ | SSIM ↑ | LPIPS ↓ | FID ↓ | KID[*] ↓ |
|---|---|---|---|---|---|---|---|---|---|---|
| **WildCAT3D (Ours)** | 21.58 | **0.758** | 0.131 | 24.70 | -0.001 | **13.92** | **0.439** | **0.355** | **9.871** | 0.015 |
| -warp | 18.97 | 0.670 | 0.182 | 28.71 | 0.003 | 13.45 | 0.410 | 0.379 | 9.934 | **0.014** |
| -warp-app | 17.73 | 0.625 | 0.216 | 31.99 | 0.008 | 12.63 | 0.368 | 0.422 | 12.41 | 0.026 |
| Base CAT3D | **21.89** | 0.751 | **0.127** | **21.53** | **-0.004** | 12.91 | 0.390 | 0.415 | 19.16 | 0.116 |

Table 2: **Quantitative ablation study.** We evaluate ablating key parts of our model, namely warp ("warp") and appearance components ("app"), when trained on in-the-wild data. We also report the performance of the base CAT3D without any in-the-wild finetuning. KID[*] indicates KID values multiplied by ten for readability. Best results are in **bold**.

# 5   Conclusion

We have presented the WildCAT3D framework for generalizable novel view synthesis learned from images in-the-wild. By explicitly modeling appearance variations, WildCAT3D unlocks training on abundant and permissively-licensed photo-tourism data, as well as allowing control over the global appearance conditions of generated views. Our results have shown its superior performance on NVS benchmarks and novel applications, while training on strictly fewer data sources than prior methods and more fully leveraging existing open web data capturing full scenes. Our experiments, while on a limited scale, demonstrate that modeling appearance enables learning from unfiltered, in-the-wild data, laying a foundation for web-scale NVS training.

**Limitations and Future Work.** Generated views are not guaranteed to be fully consistent, unlike methods using explicit 3D representations. While our method mitigates degradation due to training on mutually inconsistent data, some visual artifacts persist, such as mild flickering and saturation changes. Our architecture could be enhanced by incorporating a video prior, textual conditioning, automatic generation of camera poses, or explicit modeling of transient occlusions. Additional components may help to model semantic variations between images (e.g. holiday decorations) or view-dependent effects such as reflections. Our method relies on an existing depth estimation pipeline, and errors in predicted depth may propagate to novel views (see appendix). Finally, while our text-based conditioning via image retrieval may provide coarse appearance conditioning, future work could add explicit textual conditioning to allow more fine-grained editing or control over semantic content of scenes.

**Societal Impact.** While scene generation shows promise for positive applications in entertainment and education, we acknowledge the inherent risks of visual generative models to produce disinformation or undesired hallucinations. The central aim of our work is to encourage the adoption of open data for state-of-the-art NVS, while still requiring the same caution in responsible usage as existing generative methods.

# Acknowledgments and Disclosure of Funding

This work was sponsored by Meta AI. We thank Kush Jain and Keren Ganon for providing helpful feedback.

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

# A  Additional Results

## A.1  Video Qualitative Results

Please see the results viewer on our project page for additional results of WildCAT3D inference and applications on validation and test inputs. Videos include examples of vanilla WildCAT3D inference for novel view synthesis, and examples of applications (appearance-controlled generation, in-the-wild interpolation). We also provide the images used as input views. For vanilla inference, results include various hard-coded trajectories (lateral turns, zoom-outs, and NeRF-like circular paths). For each scene in the interpolation results, two interpolations are provided: one using the start view's appearance and the other using the end view's appearance throughout. For the appearance conditioned generation application, the images used for appearance embedding injection are also provided.

## A.2  Image Qualitative Results

Figures 8 and 9 illustrate further examples of our outputs on in- and out-of-distribution NVS benchmarks, showcasing our strong performance at predicting novel views from a single observation.

## A.3  MegaScenes–Only Ablation

In Table 3 we compare to training end-to-end using data in-the-wild from MegaScenes as the only source of multi-view data (rather than including CO3D and Re10K in multi-view training). While this underperforms our full model including these curated sources of multi-view data, it still achieves competitive performance overall, indicating that MegaScenes alone may be used to learn a strong prior on consistent novel views of scenes despite itself containing inconsistencies between scene views due to appearance variations.

## A.4  3D Reconstruction

While the main focus of our work is generating novel 2D views of a scene, we also show that these outputs may be used for downstream 3D reconstruction. In particular, we feed the 15 generated novel views of a scene (using $v = 16$ slots for WildCAT3D) into VGGSfM [Wang et al., 2024], a state-of-the-art feed-forward 3D reconstruction pipeline that recovers camera poses and the 3D geometry of the scene, and use reconstructed sparse scene geometry to initialize a 3D Gaussian Splatting representation Kerbl et al. [2023]. Examples are illustrated in Figure 10, showing that our model's outputs are sufficiently high-quality and consistent to recover underlying 3D scene geometry. Additionally, the recovered camera trajectories generally match the trajectories used for conditional generation (lateral, circular, and straight trajectories respectively in the examples shown).

| Method | DTU [Jensen et al., 2014] | | | | | Mip-NeRF 360 [Barron et al., 2022] | | | | |
| | PSNR ↑ | SSIM ↑ | LPIPS ↓ | FID ↓ | KID* ↓ | PSNR ↑ | SSIM ↑ | LPIPS ↓ | FID ↓ | KID* ↓ |
| --- | --- | --- | --- | --- | --- | --- | --- | --- | --- | --- |
| **WildCAT3D (Ours)** | 10.77 | 0.426 | 0.388 | 57.32 | 0.039 | 14.77 | 0.445 | 0.352 | 42.17 | 0.050 |
| MS-only | 9.679 | 0.362 | 0.453 | 70.42 | 0.080 | 13.68 | 0.405 | 0.403 | 47.56 | 0.063 |

| Method | Re10K [Zhou et al., 2018] | | | | | MegaScenes [Tung et al., 2024] | | | | |
| | PSNR ↑ | SSIM ↑ | LPIPS ↓ | FID ↓ | KID* ↓ | PSNR ↑ | SSIM ↑ | LPIPS ↓ | FID ↓ | KID* ↓ |
| --- | --- | --- | --- | --- | --- | --- | --- | --- | --- | --- |
| **WildCAT3D (Ours)** | 21.58 | 0.758 | 0.131 | 24.70 | -0.001 | 13.92 | 0.439 | 0.355 | 9.871 | 0.015 |
| MS-only | 18.83 | 0.673 | 0.188 | 29.15 | 0.005 | 13.30 | 0.415 | 0.378 | 10.03 | 0.016 |

Table 3: **MegaScenes-only ablation.** We compare our full model to using MegaScenes as the only source of multi-view data ("MS-only" above), discussed in Section A.3. KID* indicates KID values multiplied by ten for readability.

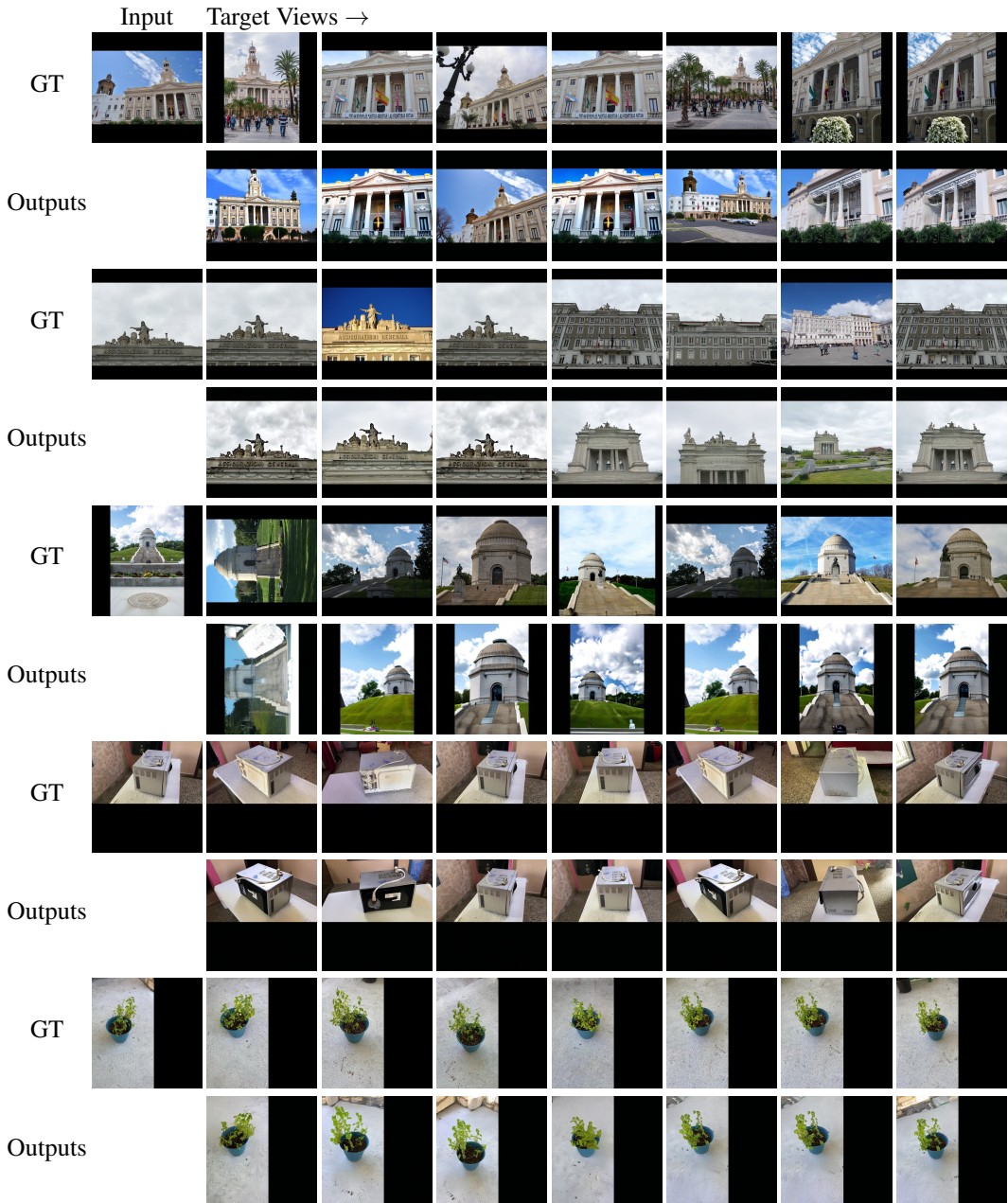

Figure 8: **In-distribution qualitative results**. Given a single input image (left), WildCAT3D can generate consistent novel views from target viewpoints (right). We here show results on test scenes from in-distribution data (MegaScenes, CO3D), OOD results can be found in Figure 9. Despite training on in-the-wild data with appearance variations, we produce views with consistent appearances (aspect ratio, global lighting, etc.) by using the appearance embedding from the input view.

## A.5 Depth Estimation Error Propogation

As our method relies on an existing depth estimation pipeline (monocular depth estimation aligned to a scene's SfM pointcloud), it may suffer from error propagation when depth is incorrectly estimated. Examples are illustrated in Figure 11, suggesting that more robust depth estimation may further improve results.

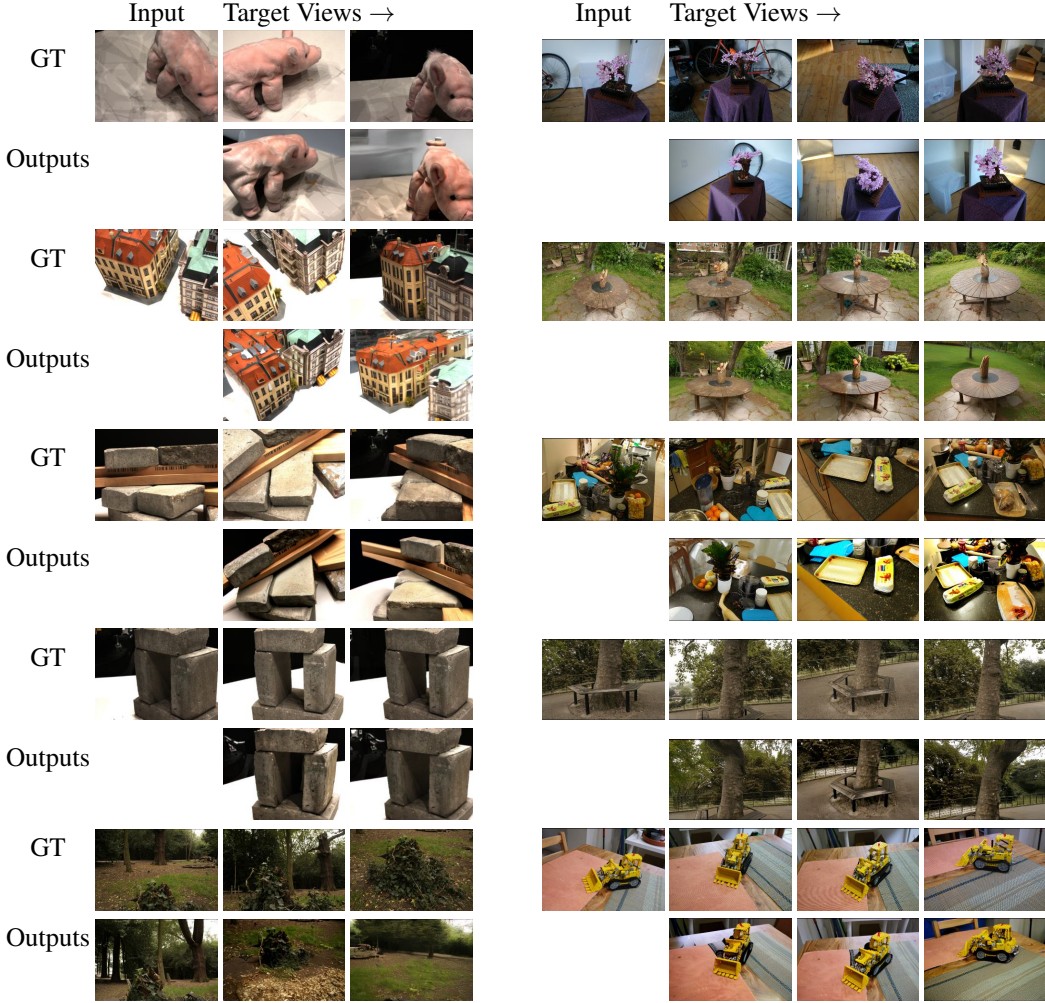

Figure 9: **Out-of-distribution (OOD) qualitative results**. Results similar to Figure 8, applied to OOD data sources (DTU, Mip-NeRF 360).

## B  Additional Implementation Details

### B.1  Image Resolution

We train on 512×512 pixel resolution ($64 \times 64$ latent resolution); images with other aspect ratios are resized so the longest edge is 512 and padded to be square. Such padding is also used by prior work and evaluation benchmarks (e.g. Gao et al. [2024], Tung et al. [2024]); unlike square cropping, it avoids losing information from the image periphery and allows for generation of images with different aspect ratios in inference. For metric calculations, we resize predictions to 256×256 resolution.

### B.2  Diffusion Process Details

Following Salimans and Ho [2022], Lin et al. [2024], we adjust noise scheduling to enforce zero terminal SNR and train with velocity prediction and loss in order to allow for generation of images with varying overall brightness levels. During inference, we generate images using CFG scale 3.

### B.3  Camera Representation Details

Following CAT3D [Gao et al., 2024], we remove a degree of ambiguity by transforming all camera poses to be relative to the first, observed pose. During training, we normalize camera scale using

Input      Sample Generated Views     3D Reconstructions (Point Cloud + Cameras)

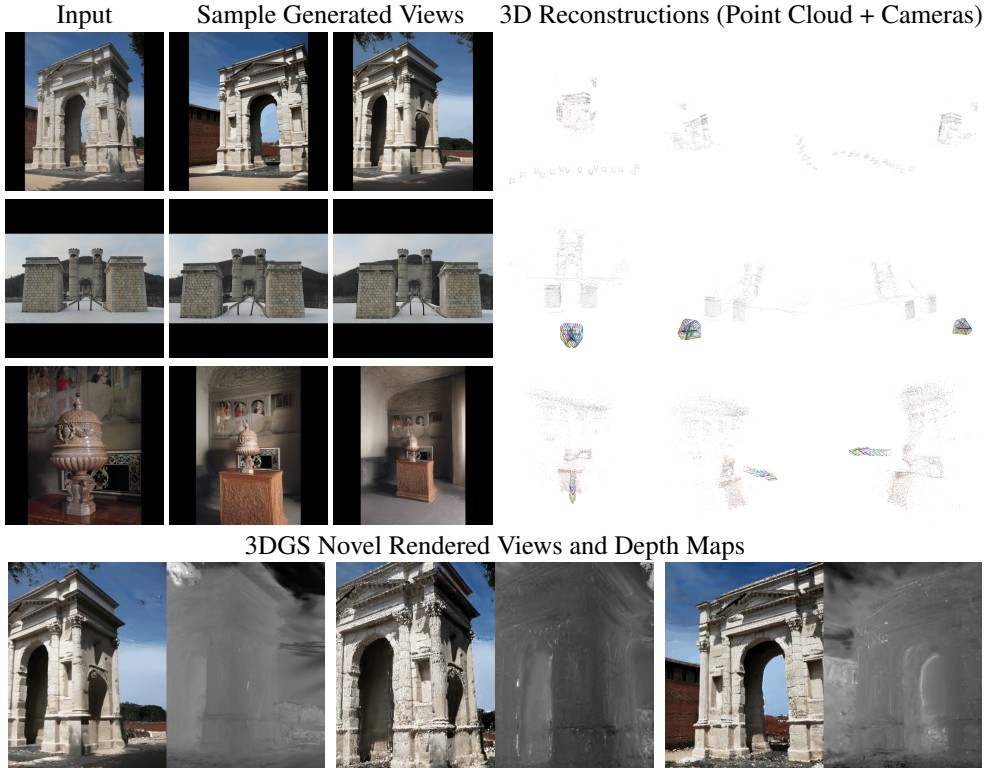

Figure 10: **3D Reconstruction from Generated Views**. We show 3D reconstruction results applied to WildCAT3D generations, as described in Appendix A.4. Reconstructed point clouds and camera positions are shown above, generally matching the expected scene geometry and camera trajectories used for generation. While two generated views are shown for conciseness, 3D reconstruction results are calculated from 15 views. We also show an example of novel views and their normalized depth maps generated using a 3D Gaussian Splatting (3DGS) representation initialized with the sparse reconstruction of the first scene.

the 10th quantile of COLMAP sparse depth visible from the first view, following ZeroNVS [Sargent et al., 2023].

We calculate Plücker raymaps as follows: given camera origin $\mathbf{o}$ and pixel $\mathbf{p}$ (vectors in world coordinates), its raw 6-dimensional coordinates are given by $(\mathbf{d}, \mathbf{o} \times \mathbf{d})$, where $\mathbf{d} = \mathbf{d} - \mathbf{o}$ is its displacement from the camera origin. As these are homogeneous coordinates describing its associated ray, we unit-normalize by dividing by the scalar factor $\sqrt{\|\mathbf{d}\|^2 + \|\mathbf{o} \times \mathbf{d}\|^2}$. This provides numerical stability by ensuring all coordinates are bounded (as extreme values may have been introduced into cameras' extrinsic matrices from translation vector scaling mentioned above). These coordinates calculated for each $64 \times 64$ spatial position provide $6 \times 64 \times 64$ channel coordinates used as input channels.

## B.4 Model Architecture

The appearance encoder module has the following architecture: It is made up of alternating convolutional layers (filter size 3, same padding) and $2 \times 2$ max pooling, with filter dimensions $16, 16, 16, 4, 2$ respectively. This converts $64 \times 64$ image VAE latents to $2 \times 2 \times 2$-dimensional embeddings, which are finally flattened to an embedding of dimension 8.

In order to be compatible with the dimensionality of added channels concatenated to latents, we add a projection layer ($1 \times 1$ convolution) to reduce the $64 \times 64 \times (2k + d + 7)$ concatenated channels to dimension $64 \times 64 \times 4$ before being input to the LDM's UNet.

| Input | Warp | Generated View | Target View |
|-------|------|----------------|-------------|

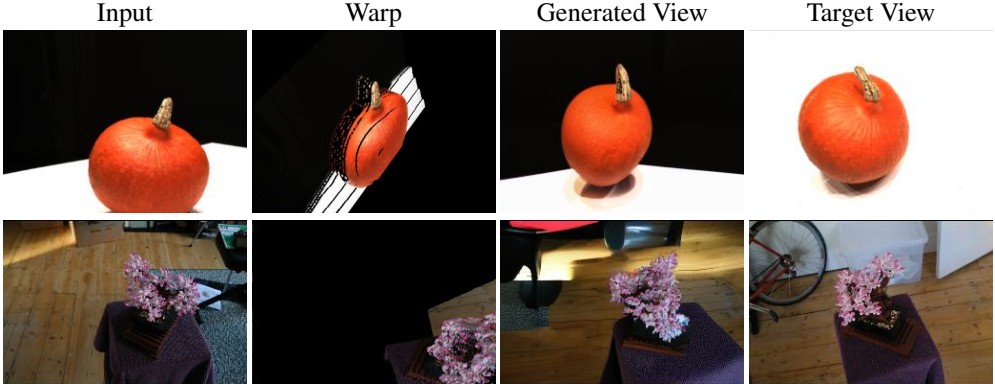

Figure 11: **Propagation of Depth Estimation Errors.** As our method relies on an existing depth estimation module to calculate warps, errors in depth estimation may propagate to novel views. This is illustrated above, as incorrect warps correspond to errors in depth estimation, which are seen to cause resulting novel generated views to deviate from the correct targets.

| | Curated Data | | | | In-the-Wild |
|-------|-------|------|------|-------|-------------|
| Method | Synth.* | ACID | CO3D | Re10K | MegaScenes |
| Zero-1-to-3 | ✓ | × | × | × | × |
| ZeroNVS | ✓ | ✓ | ✓ | ✓ | × |
| MS baseline | ✓ | ✓ | ✓ | ✓ | ✗̌ |
| WildCAT3D (Ours) | × | × | ✓ | ✓ | ✓ |
| MS-only ablation | × | × | × | × | ✓ |

Table 4: **Training data sources used**. Curated Data refers to fully consistent multiview data obtained from synthetic renderings or heavily curated videos. ✗̌ = filtered for matching time metadata and aspect ratios; MS=MegaScenes; *Objaverse(-XL)

## B.5 Training Data

Table 4 shows the sources of multi-view data used to train WildCAT3D, as well as those used in the baseline models we compare to (Zero-1-to-3 Liu et al. [2023a], ZeroNVS Sargent et al. [2023], MegaScenes baseline NVS Tung et al. [2024]). As seen in the table, our model uses strictly fewer data sources for multi-view training (neither using Objaverse [Deitke et al., 2023a,b] nor ACID [Liu et al., 2021]), and does not require aggressive filtering of in-the-wild data to avoid views with differing appearances.

Our approach successfully utilizes the full MegaScenes dataset without aggressive filtering, consisting of approximately 430K scenes and over 2M images. This is large relative to curated multi-view datasets (e.g. CO3D contains 19K videos, with 1.5M total frames). This highlights a key strength of our method – its ability to fully utilize this new, scalable source of data.

Our training data sources are all licensed under permissive licenses: Re10K and Megascenes under CC BY 4.0, and CO3D under CC BY-NC 4.0.

## B.6 Training Procedure and Compute Resources

For all model training, we use batch size 64 (where each sample in a mini-batch is itself a set of eight scene views), distributed over 32 NVIDIA A100 GPUs (using approximately 80GB of memory on each) on our internal cluster. The initial CAT3D model initialized from an open-source LDM is trained for 200K iterations, followed by 60K WildCAT3D fine-tuning iterations. End-to-end model training takes approximately one week to complete. The datasets used require several terabytes of storage, such as the 3.2 TB used to store the original images from MegaScenes, although this could be

reduced by only storing images resized to the $512 \times 512$ resolution used by our model. Preliminary experiments and each of our ablations used similar compute resources.

Compared to the leading alternative method – the MegaScenes baseline NVS model – our implementation has a larger computational footprint during training, requiring several times more GPU memory and days of training. (MegaScenes NVS was trained on 6 NVIDIA A6000 GPUs for 1-2 days.) In particular, our approach is trained to generate several (8) high-resolution (512x512) images in parallel, unlike MegaScenes NVS (which is trained to generate one 256x256 image). Our approach could be adjusted for computational efficiency in training by reducing the number of generated views and their resolution.

## B.7  Experimental Details

For evaluation benchmarks, we apply WildCAT3D with $v = 8$ input slots (matching its training procedure). As benchmarks pair one source view to a single unobserved target view, we apply WildCAT3D by grouping target views together that share the same source view. We split these into groups of seven unobserved views, padding with extra duplicated targets as needed to match the number of input slots.

To calculate generative metrics (FID, KID) on MegaScenes, we use a random 15K-item subset of the test set to make their calculation computationally feasible.

For our interpolation application (generating interpolated views between two views of a scene with differing appearances), we generate a camera trajectory as folllows: We use camera intrinsics from the first view, and interpolated extrinsics. In particular, we linearly interpolate between the camera translation vectors, and use spherical linear interpolation (slerp) to interpolate between the camera rotation matrices.

