# OpenReview forum: "WildCAT3D: Appearance-Aware Multi-View Diffusion in the Wild"
_NeurIPS.cc/2025/Conference — NeurIPS 2025 poster_

### Official Review · Reviewer_DnZH · 2025-06-16

**Clarity:** 4
**Significance:** 3
**Originality:** 4
**Rating:** 4
**Confidence:** 4

**Summary:**

This paper tries to train the CAT3D models with images captured under different conditions (lighting, occlusions). The key idea is that this pipeline could utilize a large amount of online data and has the potential to achieve a better quality if we fully utilize the online in-the-wild condition-changed multiview images. To achieve this, an additional appearance encoder is applied to let the diffusion process peak on the appearance for a better generation. To address the scale ambiguity of single-view scene generation, the method estimates an additional depth map and applies warping as another constraint in the generation. Experiments demonstrate reasonable performance improvement over the original baseline.

**Questions:**

1. About the ablation study, there could be an alternative way to describe the scene with some texts and use the texts as the appearance codes. How about the performance of such an alternative baseline, which is more straightforward?
2. Could we demonstrate 3DGS or NeRF reconstruction on the generated multiview images? Because the original CAT3D allows generating the 3D scene.
3. An alternative way is to generate the appearance-inconsistent images directly and then reconstructt them with some robust NeRF/GS methods that allow appearance changes among different images.

**Ethical Concerns:**

["NO or VERY MINOR ethics concerns only"]

**Final Justification:**

The paper explores an interesting direction, which may inspire other researchers in the field, even though the paper does not demonstrate scalability. I keep my original score with a borderline accept of this paper.

**Limitations:**

The paper mentions some limitations, but I think the main limitation is the true scalability of the proposed method. Because no experiments have been conducted to truly improve the performance with online in-the-wild multiview data.

**Quality:**

3

**Strengths And Weaknesses:**

Strength:
1. The paper is well written and clear.
2. The idea of utilizing online in-the-wild multiview images is interesting and novel. This has the potential to further scale up the training data.

Weakness:
1. The proposed method does not really demonstrate that scaling up with these in-the-wild multiview data could improve the results of CAT3D. Instead, the experiments only show that we could get some reasonable results with the proposed pipeline.
2. The resulting novel-view-synthesis quality is not so good, with visible strong flickering artifacts in the submitted videos.
3. The training still involves multiview images under the same lighting and capturing conditions. To prevent the appearance feature vector from containing too strong information about details, the proposed method applies the appearance code of the first frame as input to the network to generate other views of the same condition.

---

> ### Author Rebuttal · Authors · 2025-07-30
>
> We thank the reviewer for their constructive feedback and their positive assessment.
>
> Regarding scalability demonstration: We first note that CAT3D and other recent approaches are mostly trained on object-centric data, with poor applicability to full scenes (L28), which is a central motivation of introducing training on Internet image collections in-the-wild  [Tung et al. 2024]. In this vein, our results show meaningful improvements over the MegaScenes baseline, representing an important first step toward web-scale training on previously unusable data. In particular, the MegaScenes baseline was trained only on a small, aggressively-filtered subset of in-the-wild data, while our key contribution is showing that with appropriate conditioning mechanisms (appearance encoder + warp conditioning) we can successfully train on the full, unfiltered dataset with improved results.
>
> Regarding text-based appearance alternatives: This would be less straightforward since our training procedure learns visual appearance encodings without text supervision. Implementing text-driven appearance control would require developing additional novel components to bridge text and visual appearance spaces. Additionally, we note that our appearance module learns cues such as aspect ratio that are specific to our task, which are not directly reflected in text typically accompanying images. We will include this discussion in our revision.
>
> Regarding 3D reconstruction from generated views, we demonstrate that our method can produce coherent 3D point clouds (Appendix A.4), suggesting potential for downstream 3D reconstruction applications. These can be used to initialize a 3DGS reconstruction, which we will illustrate in our revision. Regarding the suggestion to reconstruct appearance-inconsistent views directly followed by robust volumetric reconstruction, these methods typically require scene-specific tuning and a large number of input images for successful reconstruction (more than the sparse views our method outputs in a single pass). Our method performs inference in a generic fashion on novel scenes and directly outputs mutually-consistent views without requiring additional post-processing.

---

> > ### Comment · Reviewer_DnZH · 2025-08-03
> >
> > Thanks for your response. I still have some concerns.
> > There is a large chance that we don't need such appearance-inconsistent data for training a robust NVS diffusion model.
> > I would like to see more of your insights about why we should use this kind of data and how we can improve the original model with a clear motivation. Your response with the listed experimental results is a relatively weak argument due to the small scale.

---

> > > ### Author Response · Authors · 2025-08-04
> > >
> > > Thank you for your response and continued engagement. You raise a valuable point regarding the motivation for training on appearance-inconsistent data. We see the following explicit benefits of this approach:
> > >
> > > 1. *Scale of training data:* Today's powerful models require vast amounts of training data; for example, VGGT [1] is trained on 17 datasets, and video generation models are trained on billions of web videos. Web image collections represent a tremendously large source of data, where only a small subset has been preprocessed into academic datasets (MegaDepth, MegaScenes). Being able to learn from the full breadth of web images has significant scaling potential that current methods cannot access.
> > > 2. *Scene-level data:* Most NVS datasets are object-centric (Objaverse, CO3D, DTU, etc.). The few scene-level datasets have significant limitations, such as the limited viewpoint variation in Re10K. Our approach enables training on diverse, unfiltered scene-level web data that was previously unusable.
> > > 3. *Enabling new applications:* Learning to handle appearance variations in web data naturally leads to new controlled generation capabilities, as shown by our applications in Sec 4.2.
> > >
> > > We also acknowledge that our current work represents a proof-of-concept, and that training on an even larger web scale beyond the ~430k scenes in the MegaScenes dataset would provide a stronger demonstration of scalability. While computational constraints prevent us from conducting such large-scale experiments at present, we we believe that future work may do so by leveraging our key technical contribution, as we have shown that that modeling appearance variations enables training on such unfiltered web data.
> > >
> > > We will revise our conclusion to better articulate these potential future benefits and acknowledge the current scale limitations while emphasizing the foundational nature of our technical approach.
> > >
> > > [1] Wang et al. VGGT: Visual Geometry Grounded Transformer. CVPR 2025

---

> > > > ### Comment · Reviewer_DnZH · 2025-08-06
> > > >
> > > > Thanks for your response, and I understand that this might be due to the limited computing resources.
> > > > I would keep my original scores as borderline accept.

---

### Official Review · Reviewer_tPri · 2025-07-01

**Clarity:** 3
**Significance:** 3
**Originality:** 3
**Rating:** 5
**Confidence:** 5

**Summary:**

This paper proposes an appearance-aware multi-view diffusion model that can utilize multi-view data of the same scene under different weather conditions for training. Specifically, it explicitly constructs a global appearance condition network to decouple content and appearance. During inference, an image can be designated to provide the appearance condition, enabling the generation of multi-view images with different appearances. The authors validate the proposed method through extensive ablation studies and provide many visualizations in the supplementary materials.

**Questions:**

When looking at the visualizations in the supplementary materials, I noticed that the generated multi-view images seem to have somewhat high contrast and sharpness, and the overall lighting appears unnatural. Have you explored the possible reasons behind this issue?

**Ethical Concerns:**

["NO or VERY MINOR ethics concerns only"]

**Final Justification:**

The authors’ proposed approach to decouple appearance and content is both interesting and novel, and the experimental results support their claim. Moreover, this decoupling enables training with multi-view data of different appearances, which is a significant advantage. The authors carefully addressed my concerns in the rebuttal. Although they did not conduct additional experiments due to time constraints, their explanations are reasonable. Therefore, I recommend acceptance.

**Limitations:**

The authors have already explained the limitations of their method well in the main paper.

**Quality:**

3

**Strengths And Weaknesses:**

### Strengths
1. The idea of decoupling appearance and content is very novel and impressive, and the results also validate this approach, meeting expectations.
2. Due to this decoupling, the proposed algorithm supports training with multi-view data of different appearances, and also allows specifying the appearance during inference to generate multi-view images with different appearances.
### Weaknesses
1. The authors use a fully convolutional network as the appearance model. What are the design principles of this network? For example, do factors such as its depth or number of output channels significantly influence the final results?

2. The authors use an image generator as the backbone of CAT3D. Have they tried using a video diffusion model to see its effect on the results? Additionally, what would happen if the backbone were replaced with a pure DiT structure instead of a Unet structure?

---

> ### Author Rebuttal · Authors · 2025-07-30
>
> We thank the reviewer for their constructive feedback and their positive assessment.
>
> Regarding appearance encoder design principles: The architecture requires low-dimensional output to serve as an effective information bottleneck, with gradually decreasing spatial dimensions following standard convolutional network design. Due to the substantial computational requirements of retraining the full model, extensive hyperparameter tuning of layer sizes was not feasible. Overall, we opted to design a minimal network in order to avoid additional design choices or parameters that would require testing.  We will provide more explicit discussion of the design rationale in our revision.
>
> Regarding video backbone integration: Computational constraints make this challenging for the current work, but we see this as promising future work as mentioned (L265).
>
> Regarding visual quality: As noted in our response to reviewer TYQ2, the current artifacts represent a trade-off for leveraging abundant web data, with future improvements possible through video priors or 3D distillation techniques. Addressing the reviewer’s point about high contrast and sharpness, we suspect that this relates to the use of classifier-free guidance, which is known to tend towards such artifacts in general [Sadat et al. 2024]. As we discuss (L155-167), this may be exacerbated due to appearance conditioning, and we design a strategy for mitigating this issue. We will explicitly note that this may persist to some degree as a limitation of our method and direction for future improvement.

---

> > ### Comment · Reviewer_tPri · 2025-08-05
> >
> > Thank you to the authors for their response. They addressed most of my concerns. Although they did not provide concrete experimental validation due to time and computational resource limitations, their overall explanations are reasonable. I will maintain my previous rating.

---

### Official Review · Reviewer_MpEA · 2025-07-02

**Clarity:** 3
**Significance:** 3
**Originality:** 3
**Rating:** 4
**Confidence:** 4

**Summary:**

WildCAT's key innovation is to explicitly model and disentangle appearance from 3D content. It extends the CAT3D framework by introducing two main components: 1) a generalizable Appearance Encoder that learns a low-dimensional representation of an image's global appearance, which is then used as a conditioning signal for the diffusion model, and 2) a Warp Conditioning mechanism that uses a depth-guided warp of the source view to resolve scale ambiguity and improve geometric consistency. This design allows WildCAT to train on inconsistent data while generating novel views that are consistent in both appearance and geometry at inference time. The paper demonstrates state-of-the-art performance on standard NVS benchmarks and showcases novel applications like appearance transfer and text-controlled appearance editing.

**Questions:**

1. Could you elaborate on the impact of the depth estimation quality on the warp conditioning? Have you experimented with different depth estimators or analyzed how failures in depth (e.g., on reflective surfaces or thin structures) affect the final NVS results?
2. Regarding the claim of using "strictly less data sources," could you provide a more direct comparison of the total training data volu
3. The appearance-controlled generation via CLIP retrieval is a very compelling application. Could you discuss the diversity and coverage of the database used for retrieval (MegaDepth)? Does the quality of the text-to-appearance transfer depend heavily on finding a very close match in the database, or can it generalize from moderately similar images?

**Ethical Concerns:**

["NO or VERY MINOR ethics concerns only"]

**Final Justification:**

Most questions have been clarified, and i appreciate this paper.

**Limitations:**

This is an excellent paper that makes a significant and practical contribution to the field of novel view synthesis. The problem of learning from in-the-wild data is highly relevant, and the proposed WildCAT framework is a novel, elegant, and effective solution. The paper is well-written, the experimental results are strong and comprehensive, and the demonstrated applications are impressive. The work successfully "unlocks" a new and valuable data source for training powerful generative NVS models. The weaknesses are minor and relate mostly to requests for additional details and clarifications, which do not detract from the core strength and impact of the contribution.

**Paper Formatting Concerns:**

Figure 6 caption ...results in misalignement relative to ground-truth camera poses.
The model is named "WildCAT" in the title and abstract, but is sometimes referred to as "Wildcat" (lowercase 'c') in the text and figures (e.g., Figure 2 title, "WildCAT learns..."). It would be best to make the capitalization consistent throughout the entire manuscript.

**Quality:**

4

**Strengths And Weaknesses:**

Strengths：
1. The paper tackles a significant real-world problem: how to leverage abundant but "messy" internet-scale data for NVS. By moving beyond the reliance on curated datasets, this work opens up a path towards more scalable, diverse, and generalizable NVS models, which is a crucial step for the field.
2. The proposed solution is both clever and well-motivated. The idea of an appearance encoder that allows the model to "peek" at appearance conditions during training is an elegant way to handle inconsistent data. Combining this with warp conditioning to enforce geometric consistency provides a comprehensive solution. The modifications to the classifier-free guidance to handle appearance are also a thoughtful and important detail.

weaknesses：
1. The warp conditioning module, which is shown to be critical for performance, relies on an external depth estimation model (Depth Anything). The quality and potential biases of this off-the-shelf model could significantly impact WildCAT's performance. The paper would be strengthened by a discussion on the sensitivity of the results to the accuracy of the estimated depth maps. For example, how do errors in depth estimation propagate and affect the final rendered views?
2. The paper repeatedly claims to train on "strictly less data sources than prior methods." While this may be true in terms of the number of distinct datasets used (e.g., not using Objaverse), it could be misleading. The MegaScenes dataset used for fine-tuning is itself massive. A more precise comparison of the total number of training images or training hours would provide a clearer picture of the data and computational efficiency compared to baselines like MS NVS.
3. The appearance encoder is a central component, yet its architecture is described simply as a "shallow convolutional network." Given its importance as an "information bottleneck," providing more specific details (e.g., number of layers, filter sizes, activation functions) would improve reproducibility and allow for a better understanding of its capacity and generalization capabilities.

---

> ### Author Rebuttal · Authors · 2025-07-30
>
> We thank the reviewer for their constructive feedback and their positive assessment.
>
> Regarding depth estimation sensitivity: Depth estimation errors can indeed propagate to the warp conditioning, and hence to the final views. Specifically, during our experiments, we have observed cases where incorrect depth (e.g. misplaced foreground objects in one Mip-NeRF360 scene) leads to poor alignment in novel views. We will include this failure case analysis in our revision. Note that we selected DepthAnything as it was a state-of-the-art method at the time of the submission and the same choice made by the MegaScenes baseline. Testing alternative depth estimators was computationally prohibitive given the heavy training requirements, but represents worthwhile future work.
>
> Regarding appearance encoder architecture: We provide detailed specifications in Appendix Section B.4. The design follows standard convolutional principles with gradually decreasing spatial dimensions to create a low-dimensional appearance bottleneck. We will include additional architectural motivation in our revision.
>
> Regarding data comparison and computational efficiency: Firstly, we will clarify that our use of “strictly less data sources” is intended to highlight our ability to leverage the full extent of existing, open web-scale data with less use of curated datasets which may be less scalable or have contested licensing; it is not intended to be a claim regarding sample or compute efficiency (and we will state this explicitly). We demonstrate clear improvements over the MegaScenes baseline, which used only a small, aggressively-filtered subset of available in-the-wild data. Our approach successfully utilizes the full MegaScenes dataset without aggressive filtering, consisting of approximately 430k scenes and over 2M images, showing that our method can effectively handle inconsistent appearance conditions. This is indeed large relative to curated multi-view datasets (e.g. CO3D contains 19k videos, with 1.5M total frames); we view it as a strength of our method that it is able to utilize this new, scalable source of data.  Regarding further compute and dataset details, training time is approximately one week (Appendix L86), and we will add specific comparisons of training images, dataset sizes, and computational requirements versus other methods in our revision.
>
> Regarding CLIP retrieval coverage: We use ~10k images from MegaScenes, providing the same diversity as the full dataset (tourist scenes with varying weather, lighting, and conditions). This approach primarily captures global lighting variations and weather conditions as discussed in Section 4.3. We find that this retrieval system works well in practice to capture such conditions, and we will add more explicit discussion and demonstration of these aspects.
>
> We will also correct the formatting issues and typos you identified.

---

> > ### Comment · Reviewer_MpEA · 2025-08-06
> >
> > Thanks to the authors for providing the rebuttal.
> >
> > I have read the response, as well as the responses for review comments from other reviewers. Part of the concerns have been addressed. Therefore, I am leaning towards maintaining the positive rating and suggest that the authors to incorporate all clarifications, modifications, and new experiments into the revised manuscript.

---

### Official Review · Reviewer_TYQ2 · 2025-07-02

**Clarity:** 3
**Significance:** 3
**Originality:** 2
**Rating:** 5
**Confidence:** 4

**Summary:**

The paper proposes WildCAT: A multiview diffusion model capable of NVS, that can be trained on in-the-wild image collections while also allowing for coarse appearance control over the generated images. The primary new capability introduced by the work is the ability to train a multiview diffusion model like CAT3D on inconsistent in-the-wild collections such as natural scenes for which plenty of data is available on the web. The paper enables this by proposing two main contributions: The first is an appearance encoder that extracts coarse appearance features from an input image, and the second is warp conditioning mechanism that warps pixels to their expected target locations using monocular depth estimates to better guide the image generation and resolve scale ambiguities in NVS. The core architecture of the paper builds on top an existing multiview diffusion model (CAT3D) and concatenates additional input channels for appearance and warp conditioning. The appearance embedding is a global embedding of an image; obtained through a shallow convolutional encoder applied on the image latent, and is duplicated spatially. The warp conditioning is a spatial latent map obtained by warping the input image to the target view using estimated depth from DepthAnything. Experiments show that with these improvements, a multiview diffusion model capable of NVS and coarse appearance control can be trained on inconsistent image collections.

**Questions:**

The number of additional channels required by WildCAT seems to be quite a lot more than CAT3D (d channels for the appearance, another k channels for the spatial warp). I’m curious how much this affects the size of the model and the training speed? Is it only the first layer where the capacity increases? Or did the authors increase the overall capacity of the LDM to assimilate the additional information?

**Ethical Concerns:**

["NO or VERY MINOR ethics concerns only"]

**Final Justification:**

The authors addressed my concerns in their rebuttal. I continue to hold the opinion that is an interesting first method that promises to unlock web-scale training data for NVS. Given that this is the first installment of such a method, I'm willing to cut some slack for the poor visual quality of the current results. This learnings from this work have the potential to be a very valuable contribution to the community and as such I increase my rating to 5.

**Limitations:**

There are significant temporal artifacts in the generated NVS and appearance editing results. The visual quality of the results is not very impressive.  However I do recognize and appreciate that this is the first installment of the method which introduces new capabilities and can be improved in future work. The authors address this in their work already.

**Paper Formatting Concerns:**

N/A.

**Quality:**

2

**Strengths And Weaknesses:**

*Strengths*
- The paper demonstrates interesting first results in novel view synthesis and controllable image generation, adding new capabilities to models like CAT3D. Results in the supplemental material for appearance/view interpolation show promise. This work has implications for world generation where the user can control the appearance of the generated scene to some extent.

*Weakness*
- There is significant flicker in the generated results. This makes me wonder if applying the ideas proposed in this paper over a method like Cat4D, which also has temporal priors, is a more appropriate extension of the state of the art.
- The proposed appearance and warp conditioning mechanism are reasonably straightforward and are not groundbreaking in terms of novelty.

---

> ### Author Rebuttal · Authors · 2025-07-30
>
> We thank the reviewer for their constructive feedback and their positive assessment.
>
> Regarding visual quality and temporal artifacts: We acknowledge this as a significant challenge posed by training on in-the-wild data, which contains inconsistencies not found in perfectly curated datasets. In this setting, we significantly improve over previous methods training on such data (Table 1, Fig 3), including the previously SOTA MegaScenes baseline method, which also suffer from this trade-off in order to train on abundant web data capturing full scenes. As the reviewer suggests, future improvements could include video priors (L265; such as CAT4D) or distillation from 3D priors/reconstruction to further enhance temporal consistency. We will discuss this more explicitly as a limitation of our approach and direction for future improvement in our revision.
>
> Regarding model capacity and training efficiency: We clarify that the additional input channels are handled through a lightweight 1×1 convolution projection layer (Appendix L70, Section B.4) that reduces these channels before processing. This adds very few new parameters and does not noticeably slow training or require increasing the overall LDM capacity. We will add a clarification of this important implementation detail.
>
> Regarding novelty: While individual components may seem straightforward, their combination enables a fundamentally new capability: training multiview diffusion models on inconsistent in-the-wild data. This opens access to vast amounts of previously unusable web data, representing a significant step toward scalable NVS systems. The technical challenge was not just developing these components, but making them work together effectively to handle real-world data inconsistencies that break existing methods.

---

> > ### Comment · Reviewer_TYQ2 · 2025-08-05
> >
> > Thank you for the additional clarifications :) I'm satisfied with all the answers.

---

### Decision · Program_Chairs · 2025-09-17

**Decision:**

Accept (poster)

**Comment:**

This paper proposes WildCAT, which extends a multiview diffusion model like CAT3D to enable training on inconsistent, in-the-wild image collections while supporting coarse appearance control for novel view synthesis. Its key innovations include (1) a generalizable Appearance Encoder that disentangles global appearance from 3D content, and (2) a Warp Conditioning mechanism using monocular depth to resolve scale ambiguities and improve geometric consistency. Strengths include: (1) novel decoupling of appearance and 3D geometry, enabling training on in-the-wild data; (2) the proposed appearance encoder and warp conditioning that improve view consistency; and (3) state-of-the-art performance on NVS with flexible appearance control, advancing scalable 3D content generation.

Reviewers noted that while the empirical results are compelling, the technical innovations are incremental, and the quality of generated data has room for improvement. Despite these limitations, all reviewers and the AC agree that the paper’s contributions outweigh its weaknesses. We recommend acceptance and encourage the authors to address all reviewers' comments in their manuscript revisions.